# Is Anterior Plating Superior to the Bilateral Use of Retrograde Transpubic Screws for Treatment of Straddle Pelvic Ring Fractures? A Biomechanical Investigation

**DOI:** 10.3390/jcm10215049

**Published:** 2021-10-28

**Authors:** Moritz F. Lodde, J. Christoph Katthagen, Clemens O. Schopper, Ivan Zderic, R. Geoff Richards, Boyko Gueorguiev, Michael J. Raschke, René Hartensuer

**Affiliations:** 1AO Research Institute Davos, Clavadelerstrasse 8, 7270 Davos, Switzerland; clemens.schopper@hotmail.com (C.O.S.); ivan.zderic@aofoundation.org (I.Z.); geoff.richards@aofoundation.org (R.G.R.); boyko.gueorguiev@aofoundation.org (B.G.); 2Department for Trauma, Hand and Reconstructive Surgery, University Hospital, Albert-Schweitzer-Campus 1, Building W1, Waldeyerstraße 1, 48149 Münster, Germany; christoph.katthagen@ukmuenster.de (J.C.K.); michael.raschke@ukmuenster.de (M.J.R.); hartensuer@uni-muenster.de (R.H.)

**Keywords:** straddle fractures, pelvic ring injury, biomechanics, anterior pelvic plating, retrograde transpubic screw

## Abstract

Background: Fractures of the four anterior pubic rami are described as “straddle fractures”. The aim of this study was to compare biomechanical anterior plating (group 1) versus the bilateral use of retrograde transpubic screws (group 2). Methods: A straddle fracture was simulated in 16 artificial pelvises. All specimens were tested under progressively increasing cyclic loading, with monitoring by means of motion tracking. Results: Axial stiffness did not differ significantly between the groups, *p* = 0.88. Fracture displacement after 1000–4000 cycles was not significantly different between the groups, *p* ≥ 0.38; however, after 5000 cycles it was significantly less in the retrograde transpubic screw group compared to the anterior plating group, *p* = 0.04. No significantly different flexural rotations were detected between the groups, *p* ≥ 0.32. Moreover, no significant differences were detected between the groups with respect to their cycles to failure and failure loads, *p* = 0.14. Conclusion: The results of this biomechanical study reveal less fracture displacement in the retrograde transpubic screw group after long-term testing with no further significant difference between anterior plating and bilateral use of retrograde transpubic screws. While the open approach using anterior plating allows for better visualization of the fracture site and open reduction, the use of bilateral retrograde transpubic screws, splinting the fracture, presents a minimally invasive and biomechanically stable technique.

## 1. Introduction

Fractures of the four anterior pubic rami of the pelvis, classified as “straddle” or “butterfly” fractures, were first described by Dunn and Morris [1]. Their typical mechanism implements anterior to posterior or lateral compression forces [2,3,4]. Pelvic ring fractures, caused by high energy trauma, often result in severe bleeding and complications because of the high-volume blood supply of the pelvis and injuries of the internal organs [5,6]. Mortality rates of 8–19% are reported due to bleeding or associated injuries [4,7,8]. Furthermore, limited mobilization is described as a long-term complication.

According to the AO/OTA classification, straddle fractures with an intact posterior pelvic ring are classified as type 61A2.3 [9]. According to the Tile classification, these fractures are classified as type A2 [10] and according to the Young and Burgess classification as type APC-II [2]. Discontinuity of the anterior pelvic ring—being important for its stability—is known to cause an asymmetric loading situation. The stability of the anterior pelvic ring is important for overall pelvic ring stability [11,12]. Biomechanically, both the superior and inferior pubic rami work as arches [13]. Internal fixation and especially retrograde screw fixation of pubic rami fractures was first described and published by Albin Lambotte [14,15,16]. Treatment of pelvic ring injuries has evolved from conservative treatment or limited external stabilization to internal fixation more recently [11]. Internal fixation came to be increasingly used in the 1980s and 1990s [17]. Percutaneous techniques, changing the treatment of pelvic ring injuries, were described for the first time in the mid-1990s [18]. In the presence of unstable pelvic ring injuries impeding mobilization, contemporary stable fixation is mandatory to reduce mortality and the complication rate [19]. Plating or intramedullary fixation are considered as preferred treatment options for pubic ramus fractures [11,20,21]. Some clinical studies demonstrate the successful application of retrograde transpubic screws for treatment of pubic ramus fractures in case of high energy and low energy trauma [22,23]. Simonian et al. report in their biomechanical study no significant difference between plating or screw fixation [24]. Furthermore, Marecek et al. [11] describe in a current study that there is no clear existing biomechanical evidence for the superiority of the one over the other.

Therefore, the aim of the present study was to investigate the biomechanical competence of anterior plating versus the bilateral use of retrograde transpubic screws for the treatment of straddle pelvic ring fractures. The working hypothesis was that the former will be biomechanically superior to the latter.

## 2. Materials and Methods

AO/OTA 61A2.3 anterior fractures of the four pubic rami were simulated in 16 artificial pelvises (AO 61A2.3, Model #LS4060, Synbone, Zizers, Switzerland) via vertical osteotomies of the superior and inferior pubic ramus set two centimetres laterally to the pubic tubercle. Consistency of the cuts was ensured by using a custom-made saw cut template.

The pelvises were assigned to two groups of eight specimens each (*n* = 8) for fixation with either one single anterior plate (group 1) or two bilateral retrograde transpubic screws (group 2).

In group 1, one 10-hole Dynamic Compression Plate (DCP, DePuy Synthes, Zuchwil, Switzerland) made of implant-grade stainless steel (316L) was used and pre-contoured to the shape of the bone to ensure optimal implant fit on the superior aspect of the two superior rami. The position of the plate was marked on each of the 8 pelvises for standardized implant positioning. The plate was then fixed with four mono-cortical 4.5 mm screws (DePuy Synthes) in the four medial holes and four bi-cortical 3.5 mm screws (DePuy Synthes) in the lateral holes, leaving two empty holes adjacent to the fracture site (Appendix A Figure A1). In group 2, predrilling of the retrograde transpubic screws was performed with a 3.5 mm drill bit. An aiming template made of polymethylmetharcylate (PMMA, SCS-Beracryl, Suter-Kunststoffe AG, Fraubrunnen, Switzerland) was used to achieve the best possible instrumentation reproducibility. Following their reduction, the fragments of the superior rami were fixated with one fully-threaded 4.5 × 70 mm retrograde transpubic titanium screw (Axomed GmbH, Freiburg, Germany) on each side (Figure 1B) according to the technical guidelines prescribed by Gänsslen et al. [8].

All screws were tightened at 1.5 Nm using an electronic torque screwdriver (PB 8320 A 0.4–2.5, PB Swiss Tools, Wasen/Bern, Switzerland). Optical markers were glued on the medial and lateral aspects of the fracture site of each ramus for optical motion tracking.

### 2.1. Biomechanical Testing

Biomechanical testing was performed on an electrodynamic test system Acumen (MTS Systems, Eden Prairie, MN, USA) equipped with a 3.0 kN load cell, in a setup simulating a two-legged stance position with applied load at the whole pelvis, as shown in Figure 2.

Standardization of the hip joint loading mechanics was performed using bilateral unipolar hemiarthroplasties which were custom-fit to a PMMA-potted acetabular cup on each side (Figure 2A). Cranially, each central body of the sacrum was fixed to an L-shaped frame with two screws plus washers—inserted through the first row of neuroforamina within the sacral body—and a PMMA cast. The L-shaped frame featured a radiolucent posterior section made of cotton laminates (Canevasite, HBW 2088, Amsler & Frey AG, Schinznach-Dorf, Switzerland) and was connected to the load cell and the machine actuator via a hinge joint, enabling free rotation around the longitudinal anatomical axis (Figure 2B). The specimens were aligned with the machine axis to apply axial compression force through the centre of the S1 vertebral body [25].

The loading protocol comprised an initial quasi-static ramp from an unloaded condition at 0 N to 50 N preload. Subsequently, the specimens were tested under progressively increasing sinusoidal cyclic loading at 2 Hz. Starting from 20 N, the peak load of each cycle was increased at a rate of 0.05 N/cycle, whereas the valley load was kept at a constant level of 20 N. The peak load was increased until a distinct failure of the bone-implant construct was observed, or the machine actuator reached 30 mm displacement.

### 2.2. Data Acquisition and Analysis

Machine data in terms of axial load and displacement were acquired from the machine controllers at 50 Hz. Based on the quasi-static ramp, axial stiffness was calculated from the ascending slope of the load-displacement curve within 20–40 N. Cycles to failure and failure load were evaluated retrospectively from the machine data with regard to the test stop criteria. Interfragmentary displacements were continuously captured in all six degrees of freedom by motion tracking (ARAMIS SRX, GOM GmbH, Braunschweig, Germany) at a rate of 50 Hz.

The measurement sensitivity of marker locations was ± 0.004 mm in the XY plane (frontal to the cameras) and along the *z*-axis (depth) [26,27]. A local coordinate system of the osteotomy was defined by its x-, y- and *z*-axis, oriented normally to the osteotomy plane or lying vertically and horizontally in it, respectively. The total fracture displacement (mm) was measured on both fracture sites as the magnitude of the corresponding three-dimensional displacement within the Cartesian coordinate system. Furthermore, the flexion between the medial and lateral sites of the fractured rami was calculated.

The magnitudes of fracture displacement and flexion were evaluated after 1000, 2000, 3000, 4000 and 5000 cycles with respect to the initial specimens’ state at cyclic test start under corresponding peak loading conditions. All parameters were evaluated for each fracture site separately, considering the two data sets as independent. Statistical analysis was performed using IBM SPSS Statistics (v.27, IBM, Armonk, NY, USA). Data was screened for normal distribution with the Shapiro-Wilk test. The Independent-Samples t-Test was applied to compare the normally distributed outcome measures. The Mann-Whitney U test was applied to compare the non-normally distributed outcome measures. Level for significance was set at 0.05 for all statistical tests.

## 3. Results

The outcome measures in the two study groups are summarized in Table 1, Table 2 and Table 3. Data for axial stiffness, cycles to failure and failure load showed a normal distribution. Data for total displacement and the relative flexural rotations were not normally distributed.

No significant statistical difference was detected between group 1 (anterior plate fixation) and group 2 (two retrograde transpubic screws) regarding axial stiffness (group 1: 5.10 ± 4.59 N/mm, group 2: 4.73 ± 4.23 N/mm), *p* = 0.88 (Table 1). Moreover, no significant differences were detected between two groups with respect to cycles to failure (group 1: 7816 ± 2450, group 2: 6058 ± 1695) and failure load (group 1: 410.81 ± 122.48 N, group 2: 322.91 ± 84.74 N) *p* = 0.14 (Figure 3 and Figure 4, Table 1).

Fracture displacement after 1000, 2000, 3000 and 4000 cycles was not significantly different between the two groups, *p* ≥ 0.38; however, after 5000 cycles it was significantly less in group 2 (median 0.19 cm and interquartile range 0.10 cm) compared to group 1 (median 0.36 cm and interquartile range 0.36 cm), *p* = 0.04 (Figure 5 and Table 2).

No significant differences were detected between the study groups regarding the relative flexural rotations (Table 3), *p* ≥ 0.32.

Furthermore, the standard deviations were significantly lower for group 2 compared to group 1, *p* = 0.005 (Figure 6).

The failure mode for all specimens of both groups was similar. The machine actuator reached 30 mm displacement with failure of the anterior and posterior pelvic ring.

## 4. Discussion

This biomechanical study evaluated the biomechanical competence of anterior plating versus bilateral use of retrograde transpubic screws for treatment of straddle pelvic ring fractures. From a biomechanical point of view, both fixation techniques were comparable with the retrograde screws being more stable in the long term. The hypothesis that plating would be superior to the use of bilateral retrograde transpubic screws was rejected.

Anterior plating was not superior to the retrograde transpubic screws fixation in terms of initial axial stiffness. No significant difference between the groups was observed in terms of fracture displacement after 1000, 2000, 3000 or 4000 cycles. Nonetheless, the retrograde transpubic screws led to significantly lower fracture displacement after 5000 cycles. No significant difference regarding the relative rotational flexion was detected. Furthermore, significantly less progression in the increase of standard deviation of both fracture displacement and flexion was noted for the retrograde transpubic screw fixation. The screw’s intramedullary splinting led to a more balanced distribution of the applied force compared to the anterior plating, resulting in less fracture displacement at the anterior pelvic ring and a more balanced strain exposure in the long term. Clinical studies are needed evaluating this biomechanical finding and comparing both fixations. Due to the bilateral use of the screws, this fixation technique had no disadvantages compared to the anterior plating with regard to rotational stability. These biomechanical findings confirm previous clinical data demonstrating the successful use of retrograde transpubic screws for minimally invasive stabilization of anterior pelvic ring fractures [23]. Previous biomechanical studies showed that retrograde transpubic screws and anterior plating are biomechanically comparable for treatment of straddle fractures [24,28]. The results of the present study correspond to their results. Furthermore, the results of the studies of Simonian et al. and McLachlin et al. [24,28] might indicate that the diameter of the retrograde transpubic screw is more important than its length. In a further biomechanical study it was shown that either one large 7.3 mm or two small fragment 3.5 mm retrograde screws were comparable for stabilization of pubic ramus fractures in human cadaveric hemipelvises [29]. Acklin et al. [30] performed a biomechanical comparison of plate fixation and retrograde screw using either a 7.3 mm cannulated screw or a 10-hole 3.5 mm reconstruction plate for fixation of osteoporotic pubic ramus fractures [30]. No significant difference in axial stiffness was detected. Again, this finding corresponds to the results for initial axial stiffness in the present study. Moreover, in contrast to the present study, the plating resulted in significantly less displacement than the retrograde transpubic screw fixation under progressive cyclic loading [30]. The deviating biomechanical results might be explained by the use of a 10-hole, 3.5 mm reconstruction plate stabilizing one pubic ramus. Furthermore, an osteoporotic bone model was used in the previous study. In straddle fractures with reduced bone quality the use of longer implants might be biomechanically advantageous due to a greater bone implant contact. However, the more extensive soft tissue dissection required for plating is a disadvantage. Despite the use of osteoporotic bone models, Acklin et al. [30] observed no implant breakage or screw loosening. This finding is in accordance with the present study.

From a clinical point of view, minimally displaced pubic ramus fractures might be treated conservatively. Conservatively treatment is recommended for type A2 fractures according to Tile [10]. Continuity of the anterior pelvic ring is mandatory for overall stability [11,12]. Surgical reconstruction leads to a more balanced stress distribution [30,31]. Particularly in complex pelvic ring fractures, the pubic ramus fracture can lead to an unstable situation of the pelvis [31]. When the displacement is larger than 1 cm, surgical treatment of the pubic ramus fractures should be performed [4]. Intramedullary fixation with retrograde transpubic screws is suggested for minimally invasive stabilization of superior pubic ramus fractures either on one or on both sides [21,23,32,33]. However, only indirect reduction is possible with the minimally invasive stabilization approach. The open procedure of plating allows for s direct view and open reduction but is associated with larger soft tissue and structural damage [11,21,32,33]. Bridging of comminuted and displaced superior pubic ramus fractures is achieved better by plate osteosynthesis. When discussing both treatment options, it is necessary to consider that complication rates comparable to fixation failure or immobilization rate are reported [34,35].

The limitations of this study are similar to those inherent to all biomechanical studies using synthetic bone. Results from biomechanical testing using synthetic bone without soft tissue, ligaments and muscle differ from those obtained using cadaveric models [36,37,38]. Nevertheless, synthetic bones represent an appropriate replacement for cadaver specimens [37,38,39]. Using a larger than 10-hole plate may have been beneficial for creating a stiffer construct. However, in previous biomechanical studies 10-hole plates were also used [24,28,30]. Furthermore, the reliability of the conducted procedures was achieved using standardized methods such as individually customized PMMA templates for osteotomizing and implantation (Appendix A Figure A2).

The use of an infra-acetabular screw described by Letournel originally for acetabular fracture [40] might increase the bone plate anchorage substantially and need to be examined in further biomechanical studies.

Our biomechanical findings might lead to the clinical treatment path that greater displaced anterior pelvic ring fractures could be better treated by plating via an open approach and lesser displaced fractures are better treated minimally invasive by using a retrograde transpubic screw. The majority of the previous biomechanical studies correspond to the findings of the present study [24,28]. Further biomechanical studies using human cadaveric specimens as well as clinical studies will have to confirm the applicability of our findings for clinical practice.

## 5. Conclusions

The results of this biomechanical study reveal less fracture displacement in the retrograde transpubic screw group after the long-term testing with no further significant difference between anterior plating and bilateral use of retrograde transpubic screws. While the open approach using anterior plating allows for better visualization of the fracture site and open reduction, the use of bilateral retrograde transpubic screws, splinting the fracture, presents a minimally invasive and biomechanically stable technique.

## Figures and Tables

**Figure 1 jcm-10-05049-f001:**
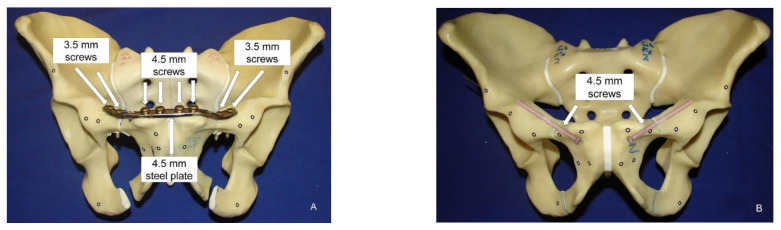
Anterior view of two specimens prepared for biomechanical testing with simulated straddle fracture of the pelvis, fixed with either an anterior plate (left, (**A**), group 1) or two bilateral retrograde transpubic screws (right, (**B**), group 2) and equipped with optical markers for motion tracking.

**Figure 2 jcm-10-05049-f002:**
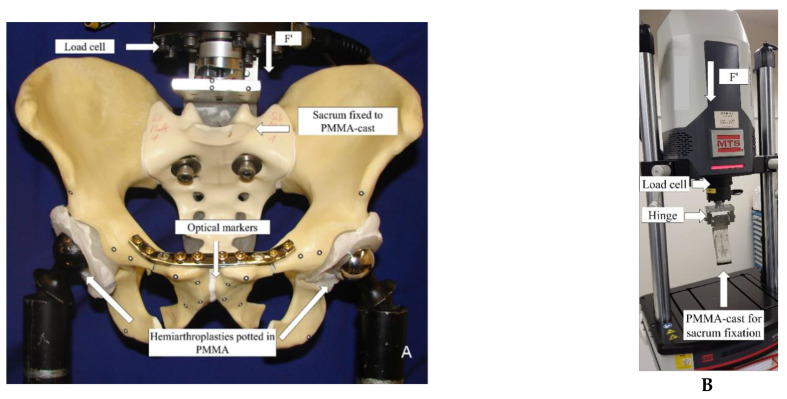
Setup with a specimen mounted for biomechanical testing in a two-legged stance position (left, (**A**)); setup without a specimen showing the PMMA cast for sacrum fixation (right, (**B**)).

**Figure 3 jcm-10-05049-f003:**
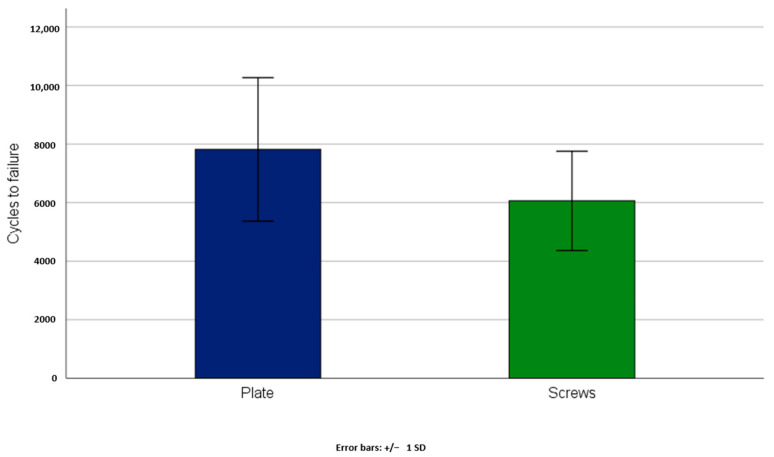
Cycles to failure in group 1 (Plate) and group 2 (Screws) presented in terms of mean value and standard deviation, with no significant difference between the groups, *p* = 0.14.

**Figure 4 jcm-10-05049-f004:**
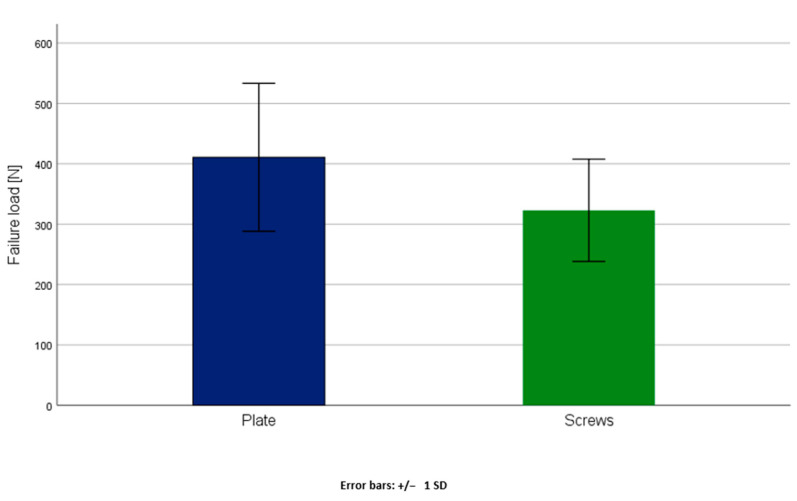
Failure load in group 1 (Plate) and group 2 (Screws) presented in terms of mean value and standard deviation, with no significant difference between the groups, *p* = 0.14.

**Figure 5 jcm-10-05049-f005:**
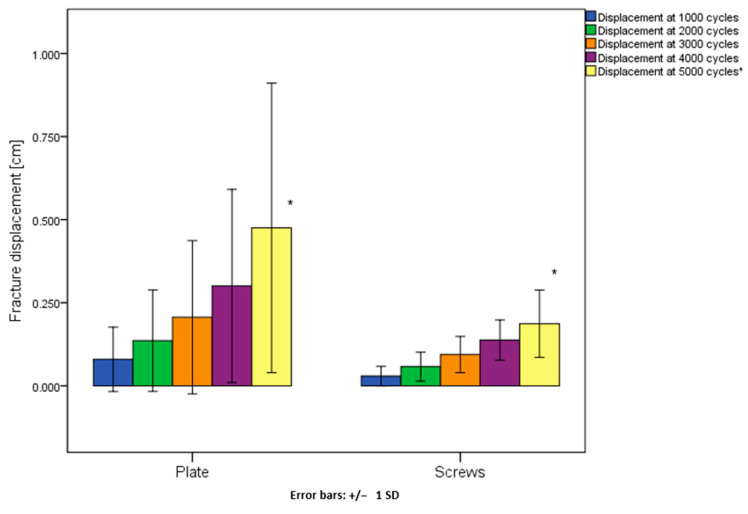
Fracture displacement after 1000, 2000, 3000, 4000 and 5000 cycles in group 1 (plate) and group 2 (screws) presented in terms of mean value and standard deviation. Significant difference between the groups after 5000 cycles is indicated with an asterisk (*).

**Figure 6 jcm-10-05049-f006:**
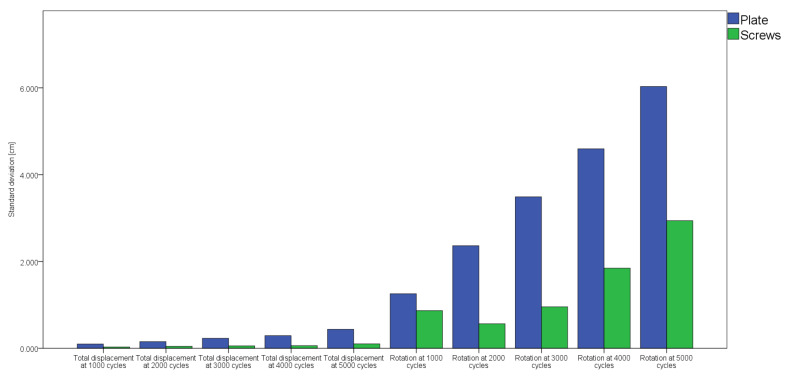
Analysis of the standard deviations revealed their significantly less progressively in increase over cycles in group 2 (screws) versus group 1 (plate) (*p* = 0.005).

**Table 1 jcm-10-05049-t001:** Axial stiffness, cycles to failure and failure load in the study groups, presented in terms of mean value and standard deviation.

Groups	Axial Stiffness [N/mm]	Cycles to Failure	Failure Load [N]
Anterior plate fixation	5.10 ± 4.59	7816 ± 2450	410.81 ± 122.48
Bilateral retrograde transpubic screws	4.73 ± 4.23	6058 ± 1695	322.91 ± 84.74

**Table 2 jcm-10-05049-t002:** Fracture displacement in the study groups after 1000, 2000, 3000, 4000 and 5000 cycles, presented in terms of median and interquartile range (the latter in brackets).

Groups	Fracture Displacement [cm]				
	at 1000 cycles	at 2000 cycles	at 3000 cycles	at 4000 cycles	at 5000 cycles
Anterior plate fixation	0.03 (0.19)	0.06 (0.29)	0.10 (0.44)	0.21 (0.52)	0.36 (0.36)
Bilateral retrograde transpubic screws	0.02 (0.02)	0.05 (0.05)	0.08 (0.06)	0.12 (0.10)	0.19 (0.10)

**Table 3 jcm-10-05049-t003:** Relative flexural rotations in the study groups after 1000, 2000, 3000, 4000 and 5000 cycles, presented in terms of median and interquartile range (the latter in brackets).

Groups	Relative Flexural Rotations [Degrees]				
	at 1000 cycles	at 2000 cycles	at 3000 cycles	at 4000 cycles	at 5000 cycles
Anterior plate fixation	0.31 (0.35)	0.42 (1.10)	1.00 (1.90)	1.73 (1.50)	2.47 (2.50)
Bilateral retrograde transpubic screws	0.26 (0.37)	0.40 (0.72)	0.77 (1.27)	1.60 (1.96)	3.40 (3.45)

## Data Availability

Data are contained within the article.

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
