# Peer review of "Is Anterior Plating Superior to the Bilateral Use of Retrograde Transpubic Screws for Treatment of Straddle Pelvic Ring Fractures? A Biomechanical Investigation"

_jcm, 2021, doi:10.3390/jcm10215049_

Round 1
Reviewer 1 Report
Dear authors. Thank you for letting me review your manuscript. It is an interesting study that can be relevant for clinical practice. My comments are largely on your English writing style, which can be improved significantly to increase readability. After processing my feedback, I would suggest to send to manuscript to a native speaker to further improvement of English writing.
Abstract
1. Line 22: I would advise to specify the groups again (group 1 plating and group 2 screws) for readability.
2. Line 24, conclusion: can you say there is no difference between plating and screwing after you write about the differences at 5000 cycles? (see also my comments on the results section).
Introduction
1. please split the sentence on mortality and limited mobilization ('Mortality rates of 8-19% are reported due to bleeding etc.', lines 38-39) It now looks as if mortality is also due to long-term complications and limited mobilization.
2. Lines 49-51 ('Internal fixation was increasingly....mid-1990s). This type of sentence does not exist in English literature. Please split the sentence to increase readability and check the manuscript for similar sentences.
3. Lines 55-59. My suggestion would be to turn the sentences around. First the clinical studies and after about the lacking biomechanical evidence. This will improve the readability and transition to the last paragraph.
4. Lines 55-56. How can I relate the part of the sentence that says there is no clear evidence of biomechanical stability to the following part that no differences were found before? (reference 22). Was this study limited by methods/numbers? Moreover, you only mention reference 22 but write about ‘some authors’.
Materials and methods
1. Lines 136-140. This already gives information about results of normally distributed data. I would suggest only to mention which test you used for normal and not-normal distribution and to not mention the outcome measures here. You give information on normality in the first section of the results.
Results
1. Figure 5 (and 6): it seems to me that there is a big difference between plating and screwing looking at the graph. From this graph it is hard to understand that there is only a significant difference after 5000 cycles. Could this be because of the scale of the Y-axis?
2. Given this result, can you still say that there is no difference between plating and screwing? (abstract, discussion). Is this difference at 5000 cycles not relevant?
3. Line 167. The ‘Start’ sign is called ‘asterisk’ in English, please change.
4. Table 2: please clarify in the table headings that the given numbers represent median and IQR (similar to table 3).
Discussion
1. Lines 183-185. Similar to my second comment in the results section. It seems to me that retrograde screws are preferable due to longer stability/less fracture displacement at all cycles. Or is this result negligible in a clinical setting? Could you provide your view on this finding?
2. Line 200: Biochemechanical study instead of studies? Or do you mean studies by Simonian and McLachlin? Then you should start a new sentence after mentioning these references. If you only want to refer to the study by Simonian, you should use ‘study’ and also place a dot behind this reference.
3. Line 201-204: This sentence is hard to follow and seems not right. Should you omit the word ‘both’? Please check.
4. Line 206: omit on the word ‘also’?
5. Lines 206-208. You refer to results of the study by Simonian but have not given information on the content of the results found in this study. Please do. I would place them before the description of results found by McLachlin (so in line 200).
6. Lines 218-220. I would place this sentence behind ….in axial stiffness was detected (line 2014) and replace ‘however’ by ‘moreover’ for readability (First similar results, than contradictive results).
7. Line 222. Conservatively and non-operative treatment are largely the same. I would suggest choosing one or another, for example: ‘…might be treated conservatively which is recommended for type A2.2….
8. Line 223: According to Tile.
9. Line 232: Try replacing the word ‘However’ in some sentences of your discussion since it repeats a lot (line 223, 229, 232) (e.g. try nonetheless, yet, but, despite etc.)
10. Lines 235-236: What kind of complications?
11. Line 238: Omit on the word ‘a’ à results from biomechanical testing
12. Line 239: ‘may be’ different or ‘is’ different?
13. Line 239. The dot after ‘models’ should be placed after the references.
14. Line 246: ‘demonstrated’ instead of ‘demonstrating’.
15. Lines 246-251. This sentence is way too long and severely decreases readability. Try splitting it up in at least three sentences. My suggestions: 1)….between plating. 2) Bilateral use of…..might lead to a clinical treatment path. 3) Within this path, largely displaced anterior….transpubic screw.
16. Lines 252-254: This sentence can be improved a lot. Since you talk about human cadaveric studies, there is no need to write about physiological bone and mineral density, this is obvious in human cadavers. My suggestion would be something like: ‘Further biomechanical studies using human cadaveric specimens as well as clinical studies will have to confirm the applicability of our findings for clinical practice’. Then you can leave out your final sentence (lines 254-255).
Conclusion
1. Line 257. You did find a significant difference and thus you cannot write this. What you can write is ‘the results of this biomechanical study revealed no large differences between….’.
2. Line 260-262. I think this conclusion can be stronger and readability can be improved. My suggestion would be something like: ‘While the open approach using anterior plating allows for better visualization of the fracture site, the use of bilateral retrograde screws is minimally invasive with the same biomechanical stability’. (anterior plating is always open reduction, so you don’t have to mention open reduction again).
Author Response
Response to Reviewer 1 Comments
Dear Editor,
thank you for your kind letter concerning our manuscript. We were happy to see the encouraging comments and wise instructions by the reviewers. Below, we comment every question and suggestion and show the changes we have made in the revised manuscript that is enclosed.
The manuscript was checked by a native English-speaking person. Please see acknowledgements.
Abstract
- Line 22: I would advise to specify the groups again (group 1 plating and group 2 screws) for readability
- correction is performed: it was significantly less in the retrograde transpubic screw group compared to the anterior plating group
- Line 24, conclusion: can you say there is no difference between plating and screwing after you write about the differences at 5000 cycles? (see also my comments on the results section).
- correction is performed: The results of this biomechanical study reveal less fracture displacement in the retrograde transpubic screw group in the long term testing with no further significant difference between anterior plating and bilateral use of retrograde transpubic screws.
Introduction
- Please split the sentence on mortality and limited mobilization ('Mortality rates of 8-19% are reported due to bleeding etc.', lines 38-39) It now looks as if mortality is also due to long-term complications and limited mobilization.
- correction is performed: Mortality rates of 8-19% are reported due to bleeding or associated injuries [4,7,8]. Furthermore, limited mobilization is described as a long-term complication.
- Lines 49-51 ('Internal fixation was increasingly....mid-1990s). This type of sentence does not exist in English literature. Please split the sentence to increase readability and check the manuscript for similar sentences.
- correction is performed: Internal fixation was increasingly used in the 1980s and 1990s [17]. Percutaneous techniques, changing the treatment of pelvic ring injuries, were described for the first time in the mid-1990s [18]. The manuscript is checked for similar sentences, and they were corrected (e.g., ll. 56-58, 199-202).
- Lines 55-59. My suggestion would be to turn the sentences around. First the clinical studies and after about the lacking biomechanical evidence. This will improve the readability and transition to the last paragraph.
- Correction is performed: Plating or intramedullary fixation are considered as preferred treatment options for pubic ramus fractures [11,20,21]. Some clinical studies demonstrate the successful application of retrograde transpubic screws for treatment of pubic ramus fractures in case of high energy and low energy trauma [22, 23]. Simonian et al. report in their biomechanical study no significant difference between plating or screw fixation [24]. Furthermore, Marecek et al.[11] describe in a current study that there is no clear exist-ing biomechanical evidence for the superiority of the one over the other.
- Lines 55-56. How can I relate the part of the sentence that says there is no clear evidence of biomechanical stability to the following part that no differences were found before? (reference 22). Was this study limited by methods/numbers? Moreover, you only mention reference 22 but write about ‘some authors’.
- Correction is performed: Plating or intramedullary fixation are considered as preferred treatment options for pubic ramus fractures [11,20,21]. Some clinical studies demonstrate the successful ap-plication of retrograde transpubic screws for treatment of pubic ramus fractures in case of high energy and low energy trauma [22, 23]. Simonian et al. report in their biomechanical study no significant difference between plating or screw fixation [24]. Furthermore, Marecek et al.[11] describe in a current study that there is no clear exist-ing biomechanical evidence for the superiority of the one over the other.
Materials and methods
- Lines 136-140. This already gives information about results of normally distributed data. I would suggest only to mention which test you used for normal and not-normal distribution and to not mention the outcome measures here. You give information on normality in the first section of the results.
- Correction is performed: Data was screened for normal distribution with the Shapiro-Wilk test. Independ-ent-Samples t-Test was applied to compare the normally distributed outcome measures. The Mann-Whitney U test was applied to compare the non-normally dis-tributed outcome measures. Level for significance was set at 0.05 for all statistical tests.
Results
- Figure 5 (and 6): it seems to me that there is a big difference between plating and screwing looking at the graph. From this graph it is hard to understand that there is only a significant difference after 5000 cycles. Could this be because of the scale of the Y-axis?
- Yes, the scale of the Y-axis was chosen in that manner due to the small difference in [cm]. However, the statistical tests showed only the significant differences as mentioned.
- Given this result, can you still say that there is no difference between plating and screwing? (abstract, discussion). Is this difference at 5000 cycles not relevant?
- Thank you very much for highlighting this point. Corrections were performed in abstract and discussion.
- Line 167. The ‘Start’ sign is called ‘asterisk’ in English, please change
- Correction is performed: asterisk
- Table 2: please clarify in the table headings that the given numbers represent median and IQR (similar to table 3).
- Corrections are performed.
Discussion
- Lines 183-185. Similar to my second comment in the results section. It seems to me that retrograde screws are preferable due to longer stability/less fracture displacement at all cycles. Or is this result negligible in a clinical setting? Could you provide your view on this finding?
- Correction is performed. However, the retrograde transpubic screws led to significantly lower fracture dis-placement after 5000 cycles. No significant difference regarding the relative rotational flexion was detected. Furthermore, significantly less progression in the increase of standard deviation of both fracture displacement and flexion was noted for the retro-grade transpubic screw fixation. The screw's intramedullary splinting led to a more balanced distribution of the applied force compared to the anterior plating, resulting in less fracture displacement at the anterior pelvic ring and a more balanced strain ex-posure in long term. There is the need for clinical studies evaluating this biomechanical finding and comparing both fixation methods prospectively.
- Line 200: Biochemechanical study instead of studies? Or do you mean studies by Simonian and McLachlin? Then you should start a new sentence after mentioning these references. If you only want to refer to the study by Simonian, you should use ‘study’ and also place a dot behind this reference.
- Correction is performed: Previous biomechanical studies showed, that retrograde transpubic screws and anterior plating are biomechanically comparable for treatment of straddle fractures [24,28]. The results of the present study correspond to their results. Furthermore, the results of the studies of Simonian et al. and McLachlin et al.[24, 28] might indicate that the diameter of the retrograde transpubic screw is more important than its length.
- Line 201-204: This sentence is hard to follow and seems not right. Should you omit the word ‘both’? Please check.
- Correction is performed: The sentence is deleted.
- Line 206: omit on the word ‘also’?
- Correction is performed.
- Lines 206-208. You refer to results of the study by Simonian but have not given information on the content of the results found in this study. Please do. I would place them before the description of results found by McLachlin (so in line 200).
- Correction is performed. Previous biomechanical studies showed, that retrograde transpubic screws and anterior plating are biomechanically comparable for treatment of straddle fractures [24,28]. The results of the present study correspond to their results. Furthermore, the results of the studies of Simonian et al. and McLachlin et al.[24, 28] might indicate that the diameter of the retrograde transpubic screw is more important than its length.
- Lines 218-220. I would place this sentence behind ….in axial stiffness was detected (line 2014) and replace ‘however’ by ‘moreover’ for readability (First similar results, than contradictive results).
- Correction is performed: Acklin et al. [30] performed a biomechanical comparison of plate fixation and retro-grade screw using either a 7.3 mm cannulated screw or a 10-hole 3.5 mm reconstruction plate for fixation of osteoporotic pubic ramus fractures [30]. No significant difference in axial stiffness was detected. Again, this finding corresponds to the results for initial axial stiffness in the present study. Moreover, in contrast to the present study, the plating resulted in significantly less displacement than the retrograde transpubic screw fixation under progressive cyclic loading [30].
- Line 222. Conservatively and non-operative treatment are largely the same. I would suggest choosing one or another, for example: ‘…might be treated conservatively which is recommended for type A2.2….
- Correction is performed. Conservatively
- Line 223: According to Tile.
- Correction is performed: According to Tile.
- Line 232: Try replacing the word ‘However’ in some sentences of your discussion since it repeats a lot (line 223, 229, 232) (e.g. try nonetheless, yet, but, despite etc.)
- Correction is performed.
- Lines 235-236: What kind of complications?
- Correction is performed. When discussing both treatment options, it is necessary to consider that comparable complication rates as fixation failure or immobilization rate are reported [34,35].
- Line 238: Omit on the word ‘a’ à results from biomechanical testing
- Correction is performed: Results from biomechanical testing using synthetic bone without soft-tissue, ligaments and muscle may be different versus use of cadaveric models [36–38].
- Line 239: ‘may be’ different or ‘is’ different?
- Correction is performed. Are different.
- Line 239. The dot after ‘models’ should be placed after the references.
- Correction is performed.
- Line 246: ‘demonstrated’ instead of ‘demonstrating’.
- Correction is performed.
- Lines 246-251. This sentence is way too long and severely decreases readability. Try splitting it up in at least three sentences. My suggestions: 1)….between plating. 2) Bilateral use of…..might lead to a clinical treatment path. 3) Within this path, largely displaced anterior….transpubic screw.
- Correction is performed.
- Lines 252-254: This sentence can be improved a lot. Since you talk about human cadaveric studies, there is no need to write about physiological bone and mineral density, this is obvious in human cadavers. My suggestion would be something like: ‘Further biomechanical studies using human cadaveric specimens as well as clinical studies will have to confirm the applicability of our findings for clinical practice’. Then you can leave out your final sentence (lines 254-255).
- Correction is performed.

Reviewer 2 Report
Thank you for this very interesting and well written biomechanical paper.
Could you comment in the discussion on the use of infra-acetabular screws and using longer plates and how this might affect your outcome. Please also comment on these fractures in osteoporotic patients where screw loosening and plate cut-out might be even more common than in good quality bone.
Author Response
Response to Reviewer 2 Comments
Dear Editor,
thank you for your kind letter concerning our manuscript. We were happy to see the encouraging comments and wise instructions by the reviewers. Below, we comment every question and suggestion and show the changes we have made in the revised manuscript that is enclosed.
Point 1: Could you comment in the discussion on the use of infra-acetabular screws and using longer plates and how this might affect your outcome?
Thank you for bringing up these important points. We discuss it now in the discussion, please see ll. 216-221. The use of longer plates is also discussed in limitations, please see ll. 243-244. Additionally, the use of longer or different screw types are already discussed, please see ll.205-210. The use of infra-acetabular screws is a very interesting suggestion which need to be examined in further studies. Please see ll. 249-251
Baumann, F., Schmitz, P., Mahr, D. et al. A guideline for placement of an infra-acetabular screw based on anatomic landmarks via an intra-pelvic approach. J Orthop Surg Res 13, 77 (2018). https://doi.org/10.1186/s13018-018-0786-1
Point 2: Please also comment on these fractures in osteoporotic patients where screw loosening and plate cut-out might be even more common than in good quality bone.
Thank you for highlighting this. We discuss this point in discussion, please see ll. 210-223.
